# LEARNING SEMILINEAR NEURAL OPERATORS: A UNIFIED RECURSIVE FRAMEWORK FOR PREDICTION AND DATA ASSIMILATION

**Ashutosh Singh**[1,*]     **Ricardo Borsoi**[2,*]     **Deniz Erdogmus**[1]     **Tales Imbiriba**[1,3]

## ABSTRACT

Recent advances in the theory of Neural Operators (NOs) have enabled fast and accurate computation of the solutions to complex systems described by partial differential equations (PDEs). Despite their great success, current NO-based solutions face important challenges when dealing with spatio-temporal PDEs over long time scales. Specifically, the current theory of NOs does not present a systematic framework to perform data assimilation and efficiently correct the evolution of PDE solutions over time based on sparsely sampled noisy measurements. In this paper, we propose a learning-based state-space approach to compute the solution operators to infinite-dimensional semilinear PDEs. Exploiting the structure of semilinear PDEs and the theory of nonlinear observers in function spaces, we develop a flexible recursive method that allows for both prediction and data assimilation by combining *prediction* and *correction* operations. The proposed framework is capable of producing fast and accurate predictions over long time horizons, dealing with irregularly sampled noisy measurements to correct the solution, and benefits from the decoupling between the spatial and temporal dynamics of this class of PDEs. We show through experiments on the Kuramoto-Sivashinsky, Navier-Stokes and Korteweg-de Vries equations that the proposed model is robust to noise and can leverage arbitrary amounts of measurements to correct its prediction over a long time horizon with little computational overhead.

## 1 INTRODUCTION

The evolution of many dynamical systems in science and engineering can be described by *partial differential equations* (PDEs), where modeled quantities are often a function of both space and time. Evolving PDEs over time can be computationally very intensive, especially for fine spatiotemporal grids and large scale systems. In this context, great effort has been recently devoted to provide neural-network (NN) approximations of integral operators as solution to such PDEs (Lu et al., 2019; Kovachki et al., 2021; Morrill et al., 2021; Kidger et al., 2020), namely, *Neural Operators* (NOs), allowing for efficiently approximating the solution to PDE systems.

Despite the efficiency and approximation power of current NOs, the literature lacks a systematic framework for *data assimilation* (Asch et al., 2016) capable of improving the estimation quality and correcting the evolving trajectories based on a small amount of noisy measurements from the system. This is very important due to advances of sensing technologies and the proliferation of data from dynamical systems in many fields such as Earth surface temperature (Jiang et al., 2023), remote sensing imaging (Weikmann et al., 2021; Borsoi et al., 2021), traffic concentration (Thodi et al., 2023), fMRI dynamics (Singh et al., 2021; Buxton, 2013) and video dynamics (Guen & Thome, 2020), to name but a few. Afshar et al. (2023) discussed the design of *observers* for the so-called *semilinar* PDEs (a class of common PDEs with a specific form of spatiotemporal dynamics) and its close connection to extended Kalman filters (Smith et al., 1962; Särkkä & Svensson, 2023). The designed observer is an important tool that can estimate the state of a system governed by some underlying

---

∗ denotes equal contribution.

[1] Department of Electrical and Computer Engineering, Northeastern University, Boston MA, 02115, USA.

[2] CNRS, CRAN, Université de Lorraine, Vandoeuvre-lès-Nancy, F-54000 Nancy, France.

[3] Institute for Experiential AI, Northeastern University, Boston MA, 02115, USA.

**Emails:** {singh.ashu,d.erdogmus,talesim}@northeastern.edu, raborsoi@gmail.com

infinite-dimensional PDEs based on measurements, showing a natural connection to data assimilation. However, standard approaches based on Kalman filters and observers require tremendous computational efforts, limiting their application to large-scale systems. Recently, a few works aimed at approximating the Kalman filter updates using NNs for video prediction (Guen & Thome, 2020), while the Kalman updates were approximated in (Revach et al., 2022). Nevertheless, a general theory and framework for data assimilation exploiting the structure of semilinear PDEs is still missing.

In this paper, we extend the theory of NOs by exploiting the observer design of semilinear PDEs and provide a systematic approach for data assimilation. The resulting method, namely *NO with Data Assimilation* (NODA), is a recursive NO approach that can be used for both estimation, when noisy measurement data is available, and prediction. To overcome the computational challenges inherent to the design and implementation of infinite-dimensional Kalman observers, we propose a data-driven NN approximation leveraging NOs (Li et al., 2020a) and learning-based Kalman estimators (Guen & Thome, 2020). Thus, NODA recursively estimates the system's states over time through *prediction* and *update* steps typical to Bayesian filtering approaches but with a small computational cost. We demonstrate through extensive simulations that NODA leads to better prediction performance when compared with other closely related NO approaches, while also being able to assimilate data with arbitrary sampling rates. This can have significant impact in practical applications including, e.g., weather and Earth surface temperature forecast (Pathak et al., 2022; Jiang et al., 2023), remote sensing imaging (Weikmann et al., 2021; Borsoi et al., 2021) and fMRI dynamics (Singh et al., 2021; Buxton, 2013). The contributions of this work can be summarized as:

- we extend the NO theory by leveraging the observer design of semilinear PDEs;
- we break the observer solution into prediction and update steps, allowing for a systematic way of performing both prediction and data assimilation, and devise a data-driven solution;
- the resulting framework can estimate solutions using arbitrary amounts of measurements.

**Related work:** Traditionally, many methods have been proposed to approximate the solution of PDEs, as can be found in (Larson & Bengzon, 2013) and (Evans, 2022). A major drawback of traditional solvers is that they are computationally taxing. A class of these methods resorts to data-driven approaches that aim at approximating the solution operator of the underlying PDE from the snapshots of the states (Kutz et al., 2016; Krstic & Smyshlyaev, 2008). Recently, NO learning has become a popular strategy for learning solution operators to a broad family of parametric PDEs (Li et al., 2020a;b; Kovachki et al., 2021; Brandstetter et al., 2022a). These methods have been further used in many applications such as for weather forecast (Pathak et al., 2022) and to track CO2 migration (Wen et al., 2022) and coastal floods (Jiang et al., 2021), among others.

NOs are also being adopted in different machine-learning paradigms. Guibas et al. (2021) present work on using a Fourier NO for token mixing in transformers. Goswami et al. (2022) motivates a physics-informed learning of NOs. Brandstetter et al. (2022b) use the symmetries of Lie groups for data augmentation. Gupta et al. (2021) proposed a multiwavelet-based operator leaning framework. Rotman et al. (2023) and Kaltenbach et al. (2023) present a semi-supervised learning approach for NO learning. Hao et al. (2023) propose the use of transformers for NO learning. Chen et al. (2023) adapts NOs for different geometries. In (Magnani et al., 2022) the authors provide a Bayesian treatment of NO for uncertainty estimation. One of the key drawbacks of these methods is that for long time scales, the solutions tend to deviate from the true trajectory. Li et al. (2022) present a Markov NO for tracking long-term trajectories of chaotic dissipative systems. However, such works have not considered the presence of noisy measurements, which is common in real-world datasets. This motivates the need for new frameworks that can address data assimilation.

Similar to the proposed work, Salvi et al. (2022) attempt to exploit the structure of the solutions of semi-linear stochastic PDEs and propose learning frameworks based on solving a fixed-point problem using ODE numerical solvers. However, the method on (Salvi et al., 2022) differs from the proposed framework as it does not address the need for correcting predictions using noisy measurement available during test time and is not recursive in nature. The capability of learning a solution operator that is flexible, recursive and robust to noisy data is essential for performing fast and accurate predictions and data assimilation over long time horizons.

Data assimilation has been well studied in the context of dynamical systems (Cheng et al., 2023; Farchi et al., 2021; Levine & Stuart, 2022; Levine, 2023). Most methods discussed in these works exploit auto-regressive formulations, which can provide good approximations of the dynamical behavior typically seen in such systems. Recently, Frion et al. (2023) extended deep dynamic mode

decomposition to learn Koopman operators through data assimilation. In the case of infinite dimensional systems, Afshar et al. (2023) discusses the observer design of semilinear PDEs and its close relationship with infinite-dimensional extended Kalman filter, which is closely connected to data assimilation methods. We leverage these principles to propose NODA as it will become clear in the remainder of this paper.

## 2 BACKGROUND

**Learning Solution Operators to PDEs:** Many applications ranging from weather forecasting to modelling molecular dynamics involve finding the solution to PDEs in both spatial and temporal variables, herein denoted by $x \in \mathbb{R}^q$ and by $t \in \mathbb{R}$, respectively. The generic PDE considered in recent works is of the form

$$
\begin{aligned}
(L_a z)(x,t) &= f(x,t), & (x,t) &\in D, \\
z(x,t) &= 0, & (x,t) &\in \partial D.
\end{aligned}
\tag{1}
$$

Here $z \in \mathcal{Z}_{xt}$ represents the solution of the PDE that belongs to a Banach space $\mathcal{Z}_{xt} \subseteq \mathcal{L}(D, \mathbb{R}^p)$, its domain given by a bounded, open set $D \subset \mathbb{R}^q \times \mathbb{R}$ with a Dirichlet boundary condition imposed over $\partial D$, $f \in \mathcal{Z}_{xt}^*$ is a known function, $L_a : \mathcal{Z}_{xt} \to \mathcal{Z}_{xt}^*$ is a differential operator with a set of parameters $a$, and $\mathcal{Z}_{xt}^*$ is the dual space of $\mathcal{Z}_{xt}$. Recent methods in neural operator learning focus on computing a parametric mapping $\mathcal{G}_\theta : a \mapsto z$ which approximates the solution to the PDE in equation 1 based on a set of pointwise evaluations $\{a_i, z_i\}_{i=1}^N$. Such approximation is performed by minimizing the empirical risk (Kovachki et al., 2021):

$$
\min_\theta \frac{1}{N} \sum_{i=1}^N \|z_i - \mathcal{G}_\theta(a_i)\|_{\mathcal{Z}}^2,
\tag{2}
$$

where $\| \cdot \|_{\mathcal{Z}}$ denotes an appropriate norm on $\mathcal{Z}$. A closely related problem consists in learning an operator that computes a snapshot solution of the PDE at a time $t > 0$, $z(\cdot, t)$, based on some initial condition $z(\cdot, 0)$ and on a set of parameters $a$.

Different NOs were recently proposed to define the parametric mappings $\mathcal{G}_\theta$, motivated by the kernel integral formulation of the solution operator to linear PDEs (Li et al., 2020b; Kovachki et al., 2021). These are given in the form of the composition of different *layer* operators, such as:

$$
\mathcal{G}(a) = (\mathcal{Q} \circ \mathcal{W}_L \circ \cdots \circ \mathcal{W}_1 \circ \mathcal{P})(a),
\tag{3}
$$

with $\mathcal{W}_\ell$ being the operator for layer $\ell$, $\mathcal{P}$ is a lifting operator, mapping $a$ to the first hidden representation, and $\mathcal{Q}$ a projection operator, mapping the last hidden representation to the output. Different forms for the operator layers have been proposed, such as the graph and multipole graph NOs (Kovachki et al., 2021), the U-shaped NO (Rahman et al., 2023), and the DeepONets (Lu et al., 2019).

A notable example is the Fourier neural operator (FNO) layer (Li et al., 2020a). The FNO was motivated by constraining the kernel in the integral operator as a shift-invariant function (Li et al., 2020b), thereby making the integral operator equivalent to a convolution that can be implemented in the frequency domain. An FNO layer, applied to an input signal $v_\ell$ and evaluated at $(x,t) \in D$, can be represented generically as

$$
v_{\ell+1}(x,t) = \sigma\left(W v_\ell(x,t) + \mathcal{F}^{-1}(R_\phi \cdot \mathcal{F}(v_\ell))(x,t)\right),
\tag{4}
$$

where $\sigma$ is a componentwise nonlinear activation function, $W$ is a linear operator, $\mathcal{F}$ is the Fourier transform, and $R_\phi$ is the Fourier transform of the kernel, parametrized by $\phi$. The FNO can be easily discretized for form a NO layer, and be made invariant to the discretization level (Li et al., 2020a).

**Semilinear PDEs:** Many applications involve PDEs which exhibit a very specific spatiotemporal structure, where the time variable has linear dynamics, while the spatial variables can be governed by more complex and possibly highly nonlinear equations (Curtain & Zwart, 2020). This specific spatiotemporal decoupling, in the so-called *semi-linear form*, is represented as (Rankin, 1993):

$$
\frac{\partial z(t)}{\partial t} = A z(t) + G(z(t), t), \qquad z(0) = z_0,
\tag{5}
$$

where $z(t) \in \mathcal{Z}_x$ denotes the solution at time index $t \in [0, t_f]$, which is itself a function of the spatial variables $x \in \mathbb{R}^q$, $A : \mathcal{D}(A) \to \mathcal{Z}_x$ be a linear bounded operator with domain $\mathcal{D}(A) \subseteq \mathcal{Z}_x$ that generates continuous semi-group operator $T(t) : \mathcal{Z}_x \to \mathcal{Z}_x$ (Curtain & Zwart, 2020), and

$G : \mathcal{Z}_x \times [0, t_f] \to \mathcal{Z}_x$ is an operator which is strongly continuous on $[0, t_f]$ and possibly non-linear in $\mathcal{Z}_x$, satisfying $G(0, t) = 0$. The operator $T(t)$ is closely linked to the temporal evolution of the PDE solution $z(t)$, as will become clearer in the following section. Note that $z(t)$ is still a function of the spatial variables $x$. Examples of PDEs with this structure include the Navier-Stokes and the Kuramoto-Sivashinsky equations, which can be seen in Section 4.

## 3 PREDICTION AND DATA ASSIMILATION WITH RECURSIVE NOS

In this section, we will present the proposed framework for a recursive method to handle both prediction of the PDE solution as well as data assimilation. Although semilinear PDEs can describe the time evolution of many infinite dimensional systems, in many practical applications such as in weather forecast (Pathak et al., 2022) or tracking of coastal floods or CO2 migration (Wen et al., 2022; Jiang et al., 2021), one does not have direct access to measures or discrete snapshots of the true solution $z(t)$. Instead, we only measure a noisy or degraded version thereof in a finite-dimensional Euclidean space, which we denote by $y(t) \in \mathbb{R}^p$. Thus, we consider the following model:

$$\frac{\partial z(t)}{\partial t} = Az(t) + G(z(t), t) + \omega(t) \,, \tag{6}$$

$$z(0) = z_0 \,, \tag{7}$$

$$y(t) = Cz(t) + \eta(t) \,. \tag{8}$$

Note that, besides a dynamical evolution based on a semilinear PDE in equations (6) and (7), the second part of the model in equation 8 describes how the measurements are generated from the true solution (or *states*) $z(t)$, with $\eta(t) \in \mathbb{R}^p$ representing the measurement noise and $C \in \mathcal{L}(\mathcal{Z}_x, \mathbb{R}^p)$ being a mapping from the solution space to the measurement space. $\omega(t) \in \mathcal{Z}_x$ is an unknown perturbation signal which describes uncertain knowledge of the system dynamics.

As mentioned previously, in many applications we only observe $y(t)$ on a discrete set of (possibly sparse, intermittent) time samples $\mathcal{T} \subset [0, t_f]$ in an interval of length $t_f$. Given an initial condition/initialization $z_0$ and the set of discrete measurements $\{y(t_k) : t_k \in \mathcal{T}\}$, our goal is to learn an *online* model capable of recovering/estimating the solution $z(t)$ *recursively* over time, using only present or past measurements $\{y(\tau) : \tau \leq t, \tau \in \mathcal{T}\}$ at every time $t$. This entails the need for a flexible method, capable of performing *prediction* in order to recover $z(t)$ at time instants when no measurements are available (using only the past estimates of $z(\tau)$ for $\tau < t$, thus dealing with missing data), and to *correct* the solution with the measured data $y(t)$ on the small number of time instants when it is observed, performing data assimilation.

In the following, we will first investigate the particular form of the solutions to the PDE in (6)–(8) and of observer designs that can provide a consistent estimate of its solution $z(t)$. This will be paramount to support the data-driven solution proposed thereafter.

### 3.1 OBSERVER DESIGN FOR SYSTEM WITH SEMILINEAR PDES

The specific structure of the semi-linear PDE allows us to write the solution in general form under mild conditions. Thus, assuming that the nonlinear operator $G$ admits a Fréchet derivative that is globally bounded as well as locally Lipschitz, uniformly in time, there exists a solution $z(t) \in \mathcal{Z}_x$ to equation 5, for a time interval $0 \leq t \leq t_f$, that can be written as (Weissler, 1979; Rankin, 1993)

$$z(t) = T(t)z_0 + \int_0^t T(t - s)G(z(s), s)ds \,. \tag{9}$$

When considering the availability of measurements $y(t)$ as in equation 8, these can be used to correct the state evolution of the system by designing an *observer*. This is performed by designing a PDE that contains a copy of the original system's dynamics with the addition of a correction term which is based on the error between the predicted and actual measurements, hereby denoted by $C\hat{z}(t) - y(t)$, by means of an observer operator gain (Afshar et al., 2023). For equations (6)–(8), the general form of the observer is given by

$$\frac{\partial \hat{z}(t)}{\partial t} = A\hat{z}(t) + G(\hat{z}(t), t) + K(t)\big[y(t) - C\hat{z}(t)\big] \,, \qquad \hat{z}(0) = \hat{z}_0 \,, \tag{10}$$

where $K(t) : \mathbb{R}^p \to \mathcal{Z}_x$ is the *observer gain*, mapping the error in the measurement space to the PDE dynamics, and $\hat{z}(t)$ is the observer solution.

By solving a nonlinear infinite-dimensional Riccati equation, one can design an operator $K(t)$ based on the PDE in equation 10 that can compute the solution $\hat{z}(t)$ over time, such that under the absence of noise and disturbances, and under mild additional conditions, the solution satisfies $\|z(t) - \hat{z}(t)\|_{\mathcal{Z}_x} \to 0$ as $t \to \infty$ for sufficiently small initial errors (see Theorem 5.1 in (Afshar et al., 2023)). Furthermore, the authors show that under bounded disturbances $\omega(t)$ and $\eta(t)$, the estimation error can also be bounded for all $t$ (see Corollary 5.2 in (Afshar et al., 2023)). Similarly to the previous case, when $K(t)$ is strongly continuous the analytical form of a solution $\hat{z}(t)$ to (10) is given by:

$$\hat{z}(t) = T(t)\hat{z}_0 + \int_0^t T(t-s)\Big[G(\hat{z}(s), s) + K(s)[y(s) - C\hat{z}(s)]\Big]ds \,. \tag{11}$$

The observer solution, however, is challenging to implement. First, the numerical solution to the infinite-dimensional Riccati equation and computation of equation 11 is difficult and computationally expensive. Moreover, it requires measurements $y(t)$ to be available at all time instants $t \in [0, t_f]$, and cannot handle missing data or perform prediction. These challenges will be addressed in the next section by the learning-based solution.

## 3.2 Learning-based Recursive Prediction-Correction NO

In this section, we provide a fully data-driven framework to compute a recursive solution to (6)–(8) which can perform prediction and handle missing data. First, we discretize the observer solution in the time domain. By rewriting the solution in equation 11, we can use it to represent the evolution of the estimate $\hat{z}(t)$ between any two discrete time instants, $t_{k-1}$ and $t_k$ (for convenience, we assume the discretization timesteps $h_k = t_k - t_{k-1}$ to be approximately constant with $k$). Moreover, it is possible to rewrite this solution in such a form that we can identify two terms, *prediction* and *correction* (see Appendix A for details), leading to:

$$\hat{z}(t_k) = \underbrace{T(t_k - t_{k-1})\hat{z}(t_{k-1}) + \int_{t_{k-1}}^{t_k} T(t_k - s)G(\hat{z}(s), s)ds}_{\text{Prediction}} + \underbrace{\int_{t_{k-1}}^{t_k} T(t_k - s)K(s)\big[y(s) - C\hat{z}(s)\big]ds}_{\text{Correction}} \,. \tag{12}$$

The prediction term depends only on the past solutions $\hat{z}(t_{k-1})$, predicting the evolution of the PDE states, whereas the correction term is based on the reconstruction error. This structure will be used in the design of the proposed learning-based solution.

Moreover, since most differential operators $A$ generate a semigroup operator $T$ which has an integral form (Gerlach, 2014), this motivates the use of an NO framework to approximate the terms in equation 12. Thus, we propose a learning-based approach that leverages the structure of (12) to perform both prediction and data assimilation in a flexible manner. Denote the estimated solution at time $t_k$ and at some spatial discretization level (typically in a regular grid) by $\hat{z}_{t_k} = \Pi_D(\hat{z}(t_k))$, where $\Pi_D$ is a discretization operator which maps from the original solution space $\mathcal{Z}_x$ to some finite-dimensional Euclidean space. Specifically, we separate the prediction and correction terms in (12), and propose architectures, described in the following, to learn each of these operations separately. The prediction operation is applied at every time instant $t_k$, generating the predicted solution $\hat{z}_{t_k}^{\text{pred}}$. Then, to adapt the method to the assimilation setting, if measurements are available (i.e., $y(t_k)$ is observed) the predicted solution is further processed by the correction operation to generate the final estimated solution $\hat{z}_{t_k}$. Otherwise, if there are no measurements, the estimated solution is set as the predicted one as $\hat{z}_{t_k} = \hat{z}_{t_k}^{\text{pred}}$. This process, which constitutes the NODA framework, is further described in the following.

**Prediction:** We approximate the first term in equation 12, which *predicts* of the solution at time $t_k$ given the estimate at time $t_{k-1}$, with a learnable NO in a residual formulation, given by:

$$\hat{z}_{t_k}^{\text{pred}} = \hat{z}_{t_{k-1}} + \mathcal{W}(\hat{z}_{t_{k-1}}) \,, \tag{13}$$

where $\mathcal{W}$ is a neural operator. Note that this operation inherits the discretization invariance from the NO $\mathcal{W}$, such as in the FNO (Li et al., 2020a). In this work, we use as $\mathcal{W}$ an FNO layer as defined in equation 4, with the Fourier transform computed only over the spatial dimension, $x$. If there are no measurements available for assimilation at time $t_k$, then we set the estimated solution as the predicted one, $\hat{z}_{t_k} = \hat{z}_{t_k}^{\text{pred}}$. If a measurement is available, however, the predicted solution is further added to the correction term as described in the following.

**Correction:** We approximate the second term in equation 12, which *corrects* the predicted solution $\hat{z}_{t_k}^{\text{pred}}$ based on the available measurement $y(t_k)$, using the following model, inspired from the observer operator form in (12) as

$$\hat{z}_{t_k} = \hat{z}_{t_k}^{\text{pred}} + K(\hat{z}_{t_k}^{\text{pred}})\big[y(t_k) - E(\hat{z}_{t_k}^{\text{pred}})\big]\,, \tag{14}$$

where $E$ is a learnable operator which approximates the (possibly unknown) measurement operator $C$ in equation 8 over a given discretization level. In this work, we parametrized $E$ using a one-hidden layer fully connected ReLU NN, which can also tackle cases where the measurement model might be nonlinear.

To parametrize the observer gain $K$, we generalized the gating architecture used in (Guen & Thome, 2020) to cope with general measurement operators $C$ and the observer design:

$$K(\hat{z}_{t_k}^{\text{pred}})[u] = \tanh\big(W_z E(\hat{z}_{t_k}^{\text{pred}}) + W_y y(t_k) + b\big) \odot \big(\hat{C}^* u\big)\,, \tag{15}$$

where $\odot$ denotes the Hadamard product and $W_z$, $W_y$ and $b$ are learnable linear operators. $\hat{C}^*$ is the conjugate operator of $\hat{C} = C \circ \Pi_D$, $\hat{C} : \hat{z}_t \mapsto y(t)$ being the discretized version of $C$. Note, however, that $\hat{C}^*$ can be a learnable linear operator in (15), particularly when $C$ is unknown.

**Learning criterion:** Given a set of training data with $S$ different realizations of trajectories of discretized solutions to the PDE (6)–(8), each generated from a different random initial conditions $z_D^{(i)}(t_0)$ and containing $N$ time samples, which we denote by $\{z_D^{(i)}(t_k), y^{(i)}(t_k)\}_{k=1}^N$, for $i = 1, \ldots, S$, NODA is learned by minimizing the following loss function:

$$\mathcal{J}(\phi) = \frac{1}{SN}\sum_{i=1}^S \sum_{k=1}^N \big\|\hat{z}_{t_k}^{(i)} - z_D^{(i)}(t_k)\big\|_2 + \frac{\lambda}{SH}\sum_{i=1}^S \sum_{k=1}^H \big\|y^{(i)}(t_k) - E\big(\hat{z}_{t_k}^{(i)}\big)\big\|_2\,, \tag{16}$$

where $z_D^{(i)}(t_k) = \Pi_D(z^{(i)}(t_k))$ denotes the the discretization of the PDE trajectory, and $\hat{z}_{t_k}^{(i)}$ denotes the recovered solution to the PDE, for the $i$-th realization. To address both the prediction and data assimilation objectives during training, NODA is initialized with $\hat{z}_{t_0}^{(i)} = z_D^{(i)}(t_0)$, then, we supply it with measurements up to a time index $t_H \le t_N$, i.e., $\{y^{(i)}(t_k)\}_{k=1}^H$, which are used to perform data assimilation in the interval $[t_1, t_H]$, after which it performs only predictions between $t_{H+1}$ and $t_N$, to generate the estimated trajectory $\{\hat{z}_{t_k}^{(i)}\}_{k=1}^N$. The loss function $\mathcal{J}(\phi)$ contains two terms. The first term measures the reconstruction of the PDE solution, while the second term measures the ability of the model to reconstruct the measurements, and the balance between these two terms is controlled by the weight $\lambda \ge 0$. The second term is particularly important when learning the operator $E$. The learnable parameters are denoted by $\phi$, and contains the parameters needed for the prediction (i.e., the NO $\mathcal{W}$) as well as the correction step (i.e., $E$ and the parameters of $K$).

## 4 EXPERIMENTS

In this section, we investigate the use of the proposed NODA framework on 1D and 2D semilinear PDEs. We present experiments with the intent of showing the efficacy of the proposed model for: **a)** Accurately predicting trajectories over long temporal horizons using only prediction steps; **b)** The efficacy of the correction step and the influence of measurement noise in the estimation performance; and **c)** The influence of the amount of sparsely measured data in the assimilation performance. For **b)** and **c)**, the snapshot measurements $y(t_k)$ are generated by synthetically adding white Gaussian noise to the clean trajectories $z_D(t_k)$, with signal-to-noise (SNR) ratios varying between 10dB and 30dB. Moreover, the noiseless case is also considered in some examples. For the third experiment, **c)**, we define as $\alpha \in [0, 1]$ the ratio between the amount of available (noisy) measurement data (sampled at random instants) and the length of the prediction horizon $\tau_{\mathrm{p}} = [t_{H+1}, t_f]$. To exploit the potential of NODA during test time, a *warm-up* procedure where prediction and update steps are performed at every time step. We define the warm-up horizon window as $\tau_{\mathrm{w}} = [t_1, t_H]$, and assume that data is available for every time step $t_k \in \tau_{\mathrm{w}}$. In all experiments, we evaluate NODA's predictive performance over the prediction horizon $\tau_{\mathrm{p}}$ for different values of $\alpha\%$ (including $\alpha = 0\%$ to evaluate only the prediction performance), of $t_f$, and for different SNRs, depending on each experiment. All the competing algorithms are initialized with the true solution after the warm-up period, $z_D(t_{H-1})$.

Table 1: Averaged RelMSE ($\times 10^3$) for prediction ($\alpha = 0\%$) on the Kuramoto-Sivashinsky equation as a function of the sequence length $t_f$ and of the SNR.

| SNR | 20 dB | | | 30 dB | | | $\infty$ | | |
|-----|-----|-----|-----|-----|-----|-----|-----|-----|-----|
| $t_f$ | 60 | 90 | 120 | 60 | 90 | 120 | 60 | 90 | 120 |
| MNO | 144±7 | 403±12 | 543±16 | 108±6 | 379±15 | 521±16 | **13**±1 | 204±8 | 410±17 |
| FNO | 298±9 | 412±12 | 715±14 | 259±10 | 324±10 | 659±13 | 223±11 | 643±19 | 892±18 |
| LSTM | 303±15 | 730±15 | 1.090±33 | 261±13 | 483±10 | 981±10 | 279±14 | 688±21 | 953±19 |
| NODA | **123**±10 | **329**±13 | **493**±10 | **63**±9 | **310**±9 | **483**±10 | 19±4 | **185**±11 | **405**±12 |

Table 2: Averaged RelMSE ($\times 10^3$) for prediction ($\alpha = 0\%$) on the Navier-Stokes equation as a function of the sequence length $t_f$ and of the SNR.

| SNR | 20 dB | | | 30 dB | | | $\infty$ | | |
|-----|-----|-----|-----|-----|-----|-----|-----|-----|-----|
| $t_f$ | 500 | 750 | 1000 | 500 | 750 | 1000 | 500 | 750 | 1000 |
| MNO | 48±7 | 72±6 | 96±12 | 39±4 | 51±7 | 87±6 | 18±3 | 22±3 | 26±5 |
| FNO | 192±10 | 294±12 | 351±14 | 189±19 | 265±21 | 317±16 | 136±11 | 177±14 | 265±16 |
| MWNO | 172±10 | 226±9 | 278±11 | 147±9 | 192±9 | 240±10 | 78±5 | 104±6 | 188±11 |
| C-LSTM | 487±15 | 522±10 | 604±18 | 397±20 | 426±17 | 481±19 | 366±11 | 417±13 | 529±11 |
| NODA | **10**±2 | **18**±4 | **32**±6 | **8**±1 | **13**±2 | **26**±4 | **7**±1 | **8**±1 | **13**±2 |

Table 3: Averaged RelMSE ($\times 10^3$) of NODA for the Navier-Stokes equation as a function of $\alpha$.

| $\alpha$ | 0% | 10% | 20% | 30% |
|-----|-----|-----|-----|-----|
| Averaged RelMSE | 26±4 | 18±4 | 13±3 | 9±3 |

We benchmark our results on a 2D PDE for the Navier-Stokes (NS) equation and on a 1D PDE for the Kuramoto-Sivashinsky (KS) equation. We also performed experiments on the Kortweg-De Vries (KdV) equation (Wazwaz, 2010), but due to space limitations the results are only included in Appendix C. These data sets represent a family of chaotic semi-linear PDEs.

**Benchmark Methods:** We compare NODA with the NO-based methods **FNO** (Li et al., 2020a), **MNO** (Li et al., 2022), and **MWNO** (Gupta et al., 2021) (for the NS), and with the auto-regressive neural network architectures **Conv-LSTM** (Shi et al., 2015) (for the NS) and **LSTM** (Chung et al., 2014) (for the KS). The MNO and LSTM-based methods were used as baselines due to their recurrent nature, which is also one of the main aspects of NODA, while FNO was selected since it is one of the main building blocks used in the NODA architecture. To evaluate the accuracy of the results of the different methods, we use the average Relative Mean Squared Error (RelMSE) to compare the accuracy of the trajectories estimated by each method, which is defined as $\text{RelMSE} = \mathbb{E}\{\sum_{t_k=t_H}^{t_f} \|z_D(t_k) - \hat{z}(t_k)\|_2^2 / \sum_{t_k=t_H}^{t_f} \|z_D(t_k)\|_2^2\}$, where $z_D(t_k)$ is the discretized ground truth sequence, $\hat{z}(t_k)$ is the sequence estimated by the method, and $\mathbb{E}\{\cdot\}$ denotes the expected value operator, which is computed over the random initializations that generate the trajectories $z_D(t_k)$. Details on how the loss function is optimized, and on architecture and hyperparameter selections are given in Appendix B.

## 4.1 DATASETS AND SETUP

**Kuramoto-Sivashinsky (KS) equation:** The one dimensional KS equation can be represented as $\frac{\partial z}{\partial t} = -z\frac{\partial z}{\partial x} - \frac{\partial^2 z}{\partial x^2} - \frac{\partial^4 z}{\partial x^4}$, where $z(t)$ is defined over the spatial domain $x \in [0, J]$, with initial condition $z(0) = z_0$. For the experiments we selected $J = 64\pi$ and periodic boundary conditions on $[0, J]$. We generate 1200 sequences with $t_f \in \{60, 90, 120\}$ seconds each and a timestep of $h = 0.25$ seconds from independent initializations are generated by randomly sampling $z_0$ according to the procedure described in (Li et al., 2022) We used 1000 sequences for training the different methods, and the remaining 200 for the evaluation of their performance. The resolution was fixed at 512 samples. To test NODA we used a warm-up period of $t_H = 40$ seconds.

**Navier-Stokes (NS) Equation:** The 2D Navier-Stokes equation represents a viscous, incompressible fluid in vorticity form on the unit torus, which can be represented as $\frac{\partial z(x,t)}{\partial t} + u(x,t)\nabla z(x,t) = v\Delta z(x,t) + f(x)$ and $\nabla \cdot u(x,t) = 0$, where $u$ is the velocity field, $z = \nabla \times u$ is the vorticity, $v = \frac{1}{Re}$, $Re > 0$ is the Reynolds number, the forcing term is $f(x) = \sin(2\pi(x + y)) + \cos(2\pi(x + y))$, and the initial condition is $z(x, 0) = z_0(x)$. Note that the NS equation can be written in semilinear form (see Section 3.3 of (Temam, 1982)).

We generated 200 independent trajectories with $t_f \in \{500, 750, 1000\}$ seconds each and a timestep of $h = 1$ second by randomly sampling $z_0$ according to the procedure described in (Li et al., 2022). We use 180 trajectories for training and evaluate on the remaining 20. We fix the resolution at $64 \times 64$ for both training and testing. We select $Re = 40$, which presents a non-turbulent case for NS, for benchmarking and comparisons. To test NODA we used a warm-up period of $t_H = 50$ seconds. Additional experiments for with higher Reynolds numbers ($Re \in \{500, 5000\}$) are available in Appendix E.

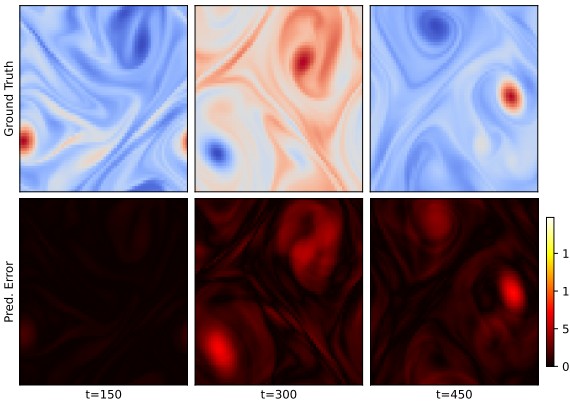

Figure 1: Samples of a realization of a true trajectory of the Navier-Stokes equation for $t \in \{150, 300, 450\}$ (**Top Row**), and elementwise error plots for the corresponding predictions by NODA (**Bottom Row**).

## 4.2 RESULTS AND DISCUSSION

**a) Prediction performance:** We evaluate the prediction of the solutions over the test interval $[t_{H+1}, t_f]$ (excluding the samples that were used as warm-up). We compare NODA with no data assimilation (i.e., using $\alpha = 0\%$) to the remaining methods, for different sequence lengths $t_f$, SNRs, and for the KS and NS equations (additional results for the KdV equation can be found in Appendix C). We compute the averaged RelMSE results, which is presented for the KS and NS equations in Tables 1 and 2, respectively.

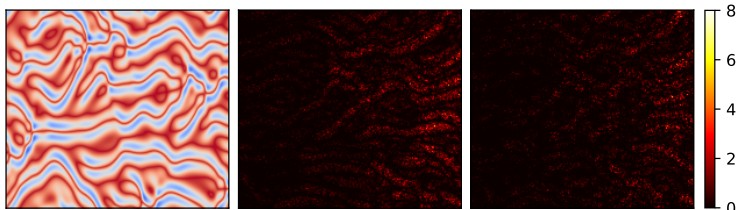

Figure 2: Prediction error for one realization of the KS equation. Warm-up with $t_H = 40$ was performed. Plots depict the prediction period $(40, 200]$ seconds. Ground truth evolution (**Left**). Error plots for NODA's solutions with $\alpha = 0\%$ (**Center**) and $\alpha = 30\%$ (**Right**).

It can be seen that the averaged errors obtained by NODA are in general significantly lower than those achieved by the competing methods, with the MNO being the second-best performing approach, followed by the MWNO (for the NS example), by the FNO and by the LSTM-based methods. This behavior is consistent across the two equations, although MNO achieves slightly better results for SNR $= 10$dB and $t_f = 60$ in the KS equation. However, we note that MNO uses the Sobolev norm and dissipativity regularizations in the training objective instead of the Euclidean norm as in the other methods. This illustrates the ability of NODA to outperform competing methods on both clean and noisy data as well as short and long rollouts. A visualization of samples of the ground truth trajectory, as well as error plots for the NODA method and the NS equation can be seen in Figure 1. The plots demonstrate the high accuracy obtained with NODA predictions, which errors that show no considerable increase between $t = 300$ and $t = 450$. Prediction results for the NS and KS equations are shown in Appendix F.

We also evaluate the influence of the amount of samples $t_H$ used in the warm-up step of NODA on its predicting performance by computing the averaged RelMSE as a function of $t_H$ for the NS equation for $\alpha = 0\%$, $t_f = 1000$ and two different SNRs. It can be seen that the error decreases considerably between $t_H = 1$ and $t_H = 150$, after which is stabilizes. This shows the benefit of the warm-up step in the performance of NODA, and also demonstrates that relatively small values of $t_H$ suffice to provide considerable performance improvements. Moreover, comparing these plots to the results the other NO-based approaches (i.e., MNO and FNO), we see that NODA provides com-

petitive performance even for $t_H = 1$ (i.e., performing predictions based on a single data snapshot), particularly for noisy data, and significantly better for larger values of $t_H$.

**b) Data assimilation performance:** We evaluate the performance of NODA while performing data assimilation in the test interval $[t_{H+1}, t_f]$ (i.e., when $\alpha > 0$). We evaluate the averaged RelMSE for the NS equation with SNR $= 30$dB and $t_f = 1000$ with different data assimilation sampling rates $\alpha \in \{0\%, 10\%, 20\%, 30\%\}$. The results are shown in Table 3, from which it can be seen that the data assimilation provides an important improvement to the estimation performance of NODA, which increases consistently with $\alpha$. Moreover, in the right panel of Figure 3 we also show the averaged RelMSE for the KS equation as a function of the both the sequence length $t_f \in [60, 140]$ and of $\alpha$, for an SNR $= 10$dB. It can be seen that the error increases only moderately with the length of the prediction interval for all tested values of $\alpha$, which indicates that NODA performs well when estimating both short and long trajectories. Moreover, the averaged RelMSE decreases consistently with $\alpha$, demonstrating the improvements of data assimilation on the results.

A realization of the ground truth and elementwise prediction errors of NODA for the KS equation, both without data assimilation and for $\alpha = 30\%$ is shown in Figure 2. It can be seen that the prediction errors tend to increase moderately with $t$, which is consistent with what was observed in Figure 3. Moreover, using data assimilation with NODA considerably reduces the prediction errors compared to the case where $\alpha = 0\%$. Additional results showcasing the data assimilation performance of NODA on the KdV equation are shown in Appendix C.

**c) Computation overhead:** To assess NODA's computational overhead in comparison with other NO-based algorithms, we measured the average time required for the prediction plus assimilation per sample for the NS ($Re = 40, \alpha = 30\%$) example. The fastest algorithm was the MNO, which took $8 \times 10^{-5}$ seconds. FNO was the slowest method, taking $2.25$ times as long as MNO, while NODA had an intermediate performance, taking $1.5$ times as long as MNO, illustrating its small computational overhead compared to MNO.

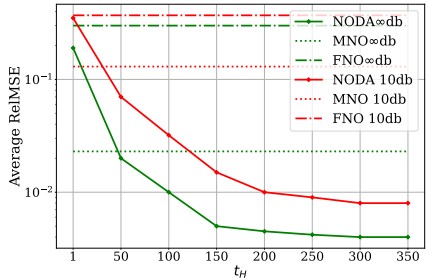 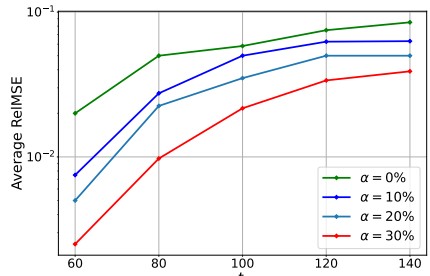

Figure 3: Average RelMSE as a function of the warm-up length $t_H$ for the NS equation for different SNRs (**Left**); and average RelMSE as function of $t_f$ for different values of $\alpha$ for the KS equation, where a warm-up period of $t_H = 40$ being used for NODA (**Right**).

## 5 CONCLUSIONS

In this work, we proposed NODA, a recursive neural operator framework for prediction and data assimilation. To do so, we extended the NO concepts by leveraging the structure of semilinear PDE systems and the correction-based state estimation paradigm from the theory of infinite dimensional observer design. As a result, NODA incorporates prediction and update steps, allowing it to perform both prediction and data assimilation depending on the availability of noisy measurement data. Extensive experiments demonstrate the capability of the proposed methodology to provide significantly more accurate estimations of the solutions of the systems even when only prediction is performed, and especially when doing data assimilation. We also show that even without the warm-up phase (i.e., when performing predictions using only a single snapshot of data), NODA solutions are still far better than the ones provided by its direct counterpart, the FNO. Nevertheless, NODA's framework is flexible and can be directly extended to incorporate other NOs and training losses proposed in the literature, such as the approach adopted by the MNO.

ACKNOWLEDGMENTS

This work was supported in part by the French National Research Agency, under grants ANR-23-CE23-0024, ANR-23-CE94-0001, and by the National Science Foundation, under grant NSF 2316420, and the National Institute of Health, NIH OT2OD030524.

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

## APPENDIX

## A    DERIVATION OF EQUATION (12)

First, let us note that as explained in (Afshar et al., 2023) equation (11) is a solution to the observer system in equation (10) when $K(t)$ is strongly continuous, due to proposition 2.4 in (Afshar et al., 2023). To improve clarity, we have made precise the need for the strong continuity of $K(t)$ for the solution in equation (11) to be valid right before equation (11) in the revised paper.

Equation (12) consists of a time-discretization of equation (11). It can be derived by using the semigroup property of $T(t)$ (Curtain & Zwart, 2020), which satisfies

$$T(t + t') = T(t)T(t')$$

for any pair $t, t'$. This way, identifying $t$ in equation (11) with the discrete time $t_k$ we can express $\hat{z}(t_k)$ as:

$$\hat{z}(t_k) = T(t_k)\hat{z}_0 + \int_0^{t_k} T(t_k - s)\Big[G(\hat{z}(s), s) + K(s)[y(s) - C\hat{z}(s)]\Big]ds \tag{17}$$

$$= T(t_k - t_{k-1} + t_{k-1})\hat{z}_0 + \int_0^{t_{k-1}} T(t_k - t_{k-1} + t_{k-1} - s)\Big[G(\hat{z}(s), s) + K(s)[y(s) - C\hat{z}(s)]\Big]ds$$
$$+ \int_{t_{k-1}}^{t_k} T(t_k - s)\Big[G(\hat{z}(s), s) + K(s)[y(s) - C\hat{z}(s)]\Big]ds \tag{18}$$

$$= T(t_k - t_{k-1})T(t_{k-1})\hat{z}_0 + T(t_k - t_{k-1})\int_0^{t_{k-1}} T(t_{k-1} - s)\Big[G(\hat{z}(s), s) + K(s)[y(s) - C\hat{z}(s)]\Big]ds$$
$$+ \int_{t_{k-1}}^{t_k} T(t_k - s)\Big[G(\hat{z}(s), s) + K(s)[y(s) - C\hat{z}(s)]\Big]ds \tag{19}$$

$$= T(t_k - t_{k-1})\underbrace{\left[T(t_{k-1})\hat{z}_0 + \int_0^{t_{k-1}} T(t_{k-1} - s)\Big[G(\hat{z}(s), s) + K(s)[y(s) - C\hat{z}(s)]\Big]ds\right]}_{=\hat{z}(t_{k-1})}$$
$$+ \int_{t_{k-1}}^{t_k} T(t_k - s)G(\hat{z}(s), s)ds + \int_{t_{k-1}}^{t_k} T(t_k - s)K(s)[y(s) - C\hat{z}(s)]ds \,, \tag{20}$$

where we used the linearity of $T(t)$. This is now in the same form as equation (12).

## B    IMPLEMENTATION DETAILS AND THE TRAINING PROCEDURE

For all the experiments, for the training of NODA we optimized the loss function $\mathcal{J}(\phi)$ using the Adam (Kingma & Ba, 2014) optimizer, using a learning rate of $10^{-4}$ with a multiplicative learning rate scheduler which decays 0.5 every 50 epochs. We train the model for a total of 300 epochs, using a batch size of 32. The weighting parameter $\lambda$ was fixed as 0.5. The backbone of NODA uses an FNO block for $\mathcal{W}$. For details on the FNO architecture, please see (Li et al., 2020a). The total number of parameters in the model was comparable to that used in the original FNO work, with a width of 64 and 20 frequencies per channel. The increase in learnable parameters for NODA compared to the FNO only depends on the choices made for the additional parameters $\{E, W_y, W_z,$

$b$} and possibly $\hat{C}^*$. In the current setting, $W_y$, $W_z$ are single linear layers, $b$ is a bias, and we choose $\hat{C}^*$ to be the identity matrix since $C$ was known a priori and equal to the identity operator in the experiments. To parametrize $E$ in a way that is flexible enough to approximate the measurement process for different data acquisition models, we considered a 2-layer fully connected neural network with ReLU activation function. Thus, in the current setting of NODA, its additional parameters constitute only light increase to the number of learnable parameters in the base FNO architecture. Hence, the training time of NODA remains comparable to the original FNO (2D+time). For the experiments on the KS equation, we chose a 1D FNO as our base model, composed of four Fourier layers, whereas for the experiments on the NS equation, we considered a 2D FNO, also with four Fourier layers. To implement the MNO (Li et al., 2022), the FNO (Li et al., 2020a) the Conv-LSTM (Shi et al., 2015), and the LSTM (Chung et al., 2014) baselines, we considered the setup described in the original papers.

For the NS equation, we train NODA using the first 500 time samples of the 180 trajectories contained in the training dataset. We choose $t_H^{\text{train}} = 300$ for training of the model on the NS dataset. This allows the model to use both the prediction and the correction terms of the loss function in the first 300 samples, while only using the prediction part for the remaining 200 snapshots. As for the KS dataset, we selected $t_H^{\text{train}}$ as 500. We trained NODA on 1000 trajectories contained in the training dataset, each containing a total of 800 snapshots. We generate the ground truth trajectories for the KS equation using the exponential time-differencing fourth-order Runge-Kutta method from (Kassam & Trefethen, 2005). To generate the trajectories for the NS equation, we used the Crank–Nicolson scheme as described in (Li et al., 2020a).

All the experiments were performed on computation nodes with Intel Xeon Gold 6132 CPUs and Nvidia Tesla V100 GPUs. Codes used to implement NODA can be found at: `https://github.com/singh17ashu/NODA-Neural-Operator-with-Data-Assimilation`

## C    Results for the Korteweg-de Vries (KdV) equation

**Korteweg-de Vries (KdV) equation:** The KdV PDE is given by

$$\frac{\partial z}{\partial t} = -z\frac{\partial z}{\partial x} - \frac{\partial^3 z}{\partial^3 x}\,,  \tag{21}$$

with $z(t) \in [0, J]$, where $J = 128$, and $F$ is a nonlinear operator. A periodic boundary condition is considered for this equation, and initial conditions $z(0) = z_0$. This is a third order semilinear PDE (Sonner, 2022). For more details on this PDE, please refer to (Wazwaz, 2010). This equation represents the propagation of waves on a fluid surface when subjected to a perturbation. We generated 1200 trajectories from random initializations $z_0$ sampled as described in (Brandstetter et al., 2022b), containing $t_f \in \{100, 150, 200\}$ seconds each, with a timestep of $h = 0.5$ seconds. The first 1000 trajectories were used for training NODA and the competing methods, while the remaining 200 were used for evaluating their performance. The resolution was fixed at 128 samples. To test NODA we used a warm-up period of $t_H = 40$ seconds.

**Benchmark Methods:** For the KdV equation, we compare NODA with the **FNO** (Li et al., 2020a) and with the **GRU** (Chung et al., 2014) methods.

The average RelMSE results of all methods for prediction ($\alpha = 0\%$) are shown in Table 4, where one can see a clear improvement on the performance of NODA when compared to the benchmark algorithms. Regarding the data assimilation performance, Figure 4 shows the Average RelMSE for different values of $t_f$ and for different values of $\alpha$. The results corroborate the findings stated in the main manuscript with the amount of available data directly impacting NODA's performance.

Plots containing samples of the ground truth solution, of NODA's estimation, and of elementwise estimation errors are shown in Figure 5, both with ($\alpha = 30\%$) and without ($\alpha = 0\%$) data assimilation. These results show that the estimation errors increase only moderately over time, and that data assimilation can significantly improve the estimation results. Different samples of the ground truth trajectories and of the corresponding predictions by NODA for $\alpha = 0\%$ are shown in Figure 6, where it is shown that NODA provides accurate predictions of the solution for different initial conditions.

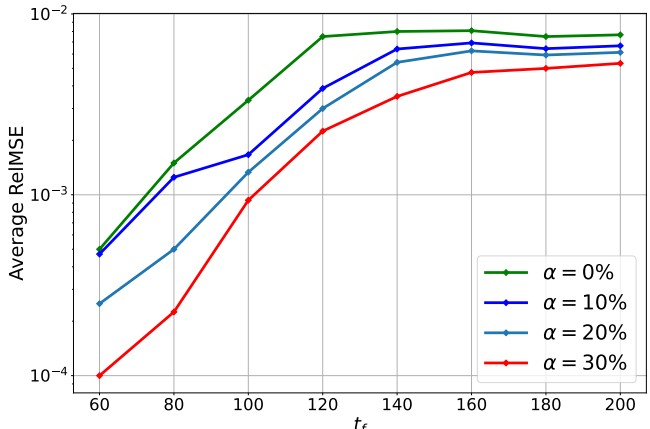

Figure 4: Average RelMSE as function of $t_f$ for different values of $\alpha$ for the KdV equation, where a warm-up period of $t_H = 40$ being used for NODA.

Table 4: Averaged RelMSE ($\times 10^3$) for prediction ($\alpha = 0\%$) on the Korteweg-de Vries equation as a function of the sequence length $t_f$ and of the SNR.

| SNR | 20 dB | | | 30 dB | | | $\infty$ | | |
|---|---|---|---|---|---|---|---|---|---|
| $t_f$ | 100 | 150 | 200 | 100 | 150 | 200 | 100 | 150 | 200 |
| FNO | 21 | 129 | 164 | 17 | 95 | 134 | 13 | 79 | 112 |
| MWNO | 18 | 107 | 149 | 15 | 88 | 114 | 10 | 59 | 92 |
| GRU | 207 | 296 | 421 | 190 | 219 | 258 | 122 | 178 | 254 |
| NODA | **14** | **89** | **102** | **12** | **76** | **92** | **9** | **48** | **61** |

## D  RESULTS FOR DIFFERENT MEASUREMENT MODELS

In this section we evaluate the performance of NODA for two different choices of the discretized measurement operator $\hat{C}$. We present results for the NS example ($R_e = 40, t_f = 1000, \text{SNR} = 30\text{dB}$), where the first choice of measurement operator is $\hat{C} = I$ (the identity operator, also considered in the main body of the paper), and $\hat{C}_{\text{rand}}$, which is a full matrix generated randomly, with each element sampled uniformly in the interval $\hat{c}_{ij} \in [0, 1]$. In Table 5 we present the averaged RelMSE results for the two choices of the measurement operator and for different choices of $\alpha$. It can be seen that the example with the random operator $\hat{C}_{\text{rand}}$ is noticeably more challenging, as evidenced by the consistently higher averaged RelMSE values. Nevertheless, the increase in the amount of data available for assimilation (i.e., $\alpha$) consistently increases the performance of NODA for both choices of measurement operator.

Table 5: Averaged RelMSE ($\times 10^3$) of NODA for the Navier-Stokes equation as a function of $\alpha$ for two different discretized measurement operators $\hat{C}$.

| $\alpha$ | 0% | 10% | 20% | 30% |
|---|---|---|---|---|
| $\hat{C} = I$ | 26 | 18 | 13 | 9 |
| $\hat{C}_{\text{rand}}$ | 42 | 39 | 33 | 25 |

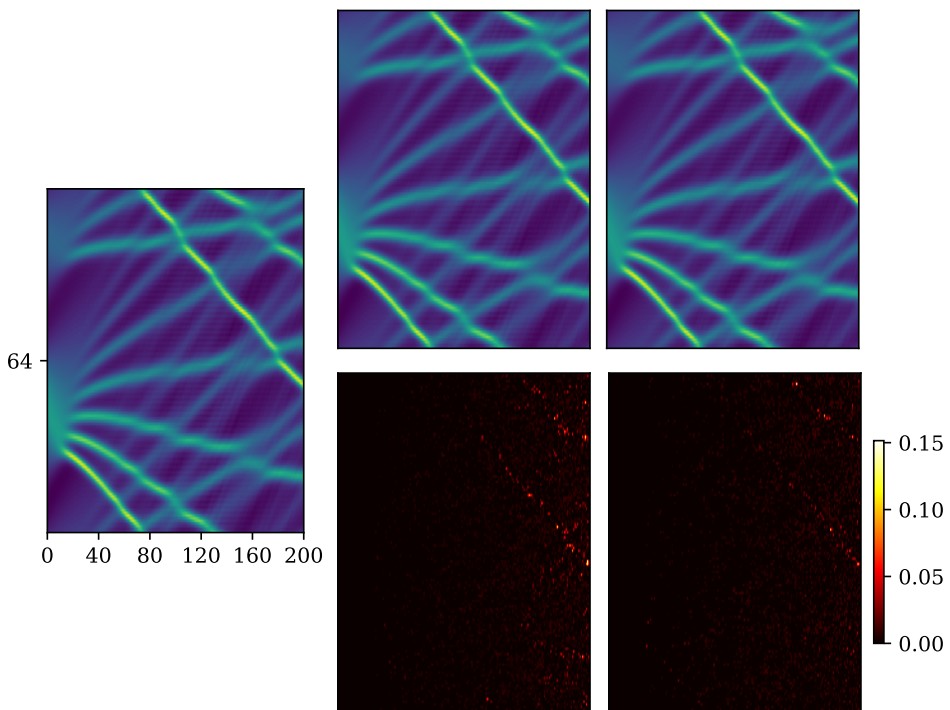

Figure 5: Sample of the predictionresults for the Korteweg-de Vries equation with $t_f = 200$ seconds. Ground truth trajectory (**Left Image**). Solution estimated by NODA with $\alpha = 0\%$ (**Top left**). Solution estimated by NODA with $\alpha = 30\%$ (**Top Right**). Elementwise error plot for NODA's estimate without data assimilation (**Bottom Left**). Elementwise error plot for NODA's estimate with data assimilation (**Bottom Right**).

Table 6: Averaged RelMSE ($\times 10^3$) of NODA, MNO and FNO for the Navier-Stokes equation as a function of $\alpha$ for $Re \in \{500, 5000\}$.

|  | $Re$ / $\alpha$ | 0% | 10% | 20% | 30% |
|---|---|---|---|---|---|
| NODA | 500 | 196 | 179 | 152 | 145 |
|  | 5000 | 258 | 226 | 202 | 189 |
| MNO | 500 | 183 | – | – | – |
|  | 5000 | 221 | – | – | – |
| FNO | 500 | 247 | – | – | – |
|  | 5000 | 310 | – | – | – |

# E    RESULTS FOR THE NAVIER-STOKES EQUATION WITH DIFFERENT VORTICITIES

In this section, we present results for NS equation with $Re \in \{500, 5000\}$, corresponding to flows with higher turbulence. We compare the results of NODA, MNO and FNO. For NODA we show the results for different values of $\alpha \in \{0\%, 10\%, 20\%, 30\%\}$, while for the remaining methods only $\alpha = 0\%$ is shown as they do not perform data assimilation. For both choices of $Re$, we considered SNR $= 30$dB, $t_H = 50$ and $t_f = 500$. The averaged RelMSE results are shown in Table 6. From these results, it can be seen that when $\alpha = 0\%$ (for prediction only), NODA outperforms FNO and has performance that is competitive with MNO, which provided the lowest average RelMSE values. However, as more data is used for assimilation (particularly for $\alpha \in \{20\%, 30\%\}$), the average RelMSE of NODA decreases significantly, becoming lower than that those provided by both MNO and FNO. This highlights the benefits of data assimilation in the performance of NODA.

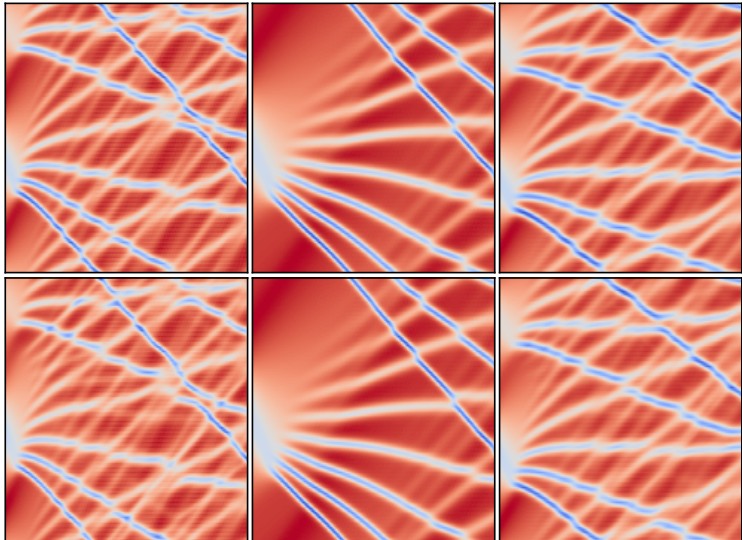

Figure 6: Sample of the prediction results for the Korteweg-de Vries equation with $t_f = 200$ seconds. Three different realisations of the ground truth trajectory generated from different initial conditions (**Top Row**). Corresponding trajectories predicted by NODA (**Bottom Row**).

## F  ADDITIONAL RESULTS ON THE NAVIER-STOKES AND KURAMOTO-SIVASHINSKY EQUATIONS

In this section, we present complementary results for the KS and NS equations. These consist of visualizations of samples of the ground truth and reconstructed trajectories plots for predictions of NODA (using $\alpha = 0\%$). These can be seen in Figures 7 and 8.

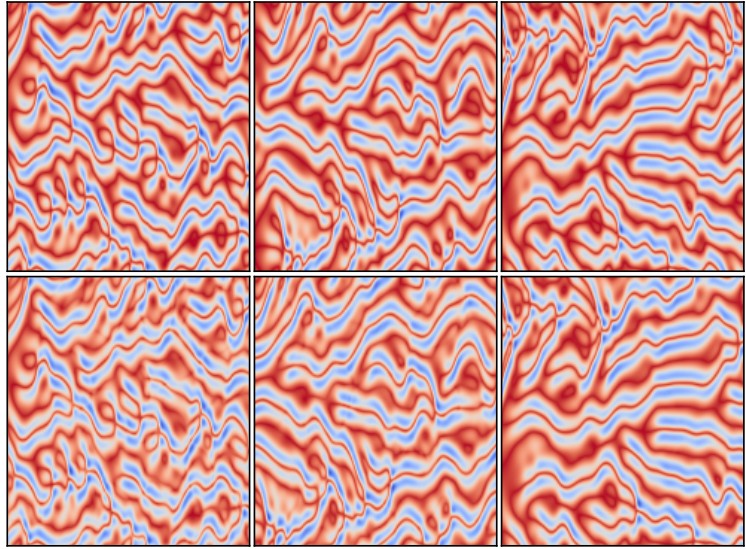

Figure 7: Different realizations of ground truth trajectories of the Kuramoto-Sivashinsky equation with $t_f = 200$ seconds (**Top Row**), and the corresponding predictions obtained by NODA without data assimilation (**Bottom Row**).

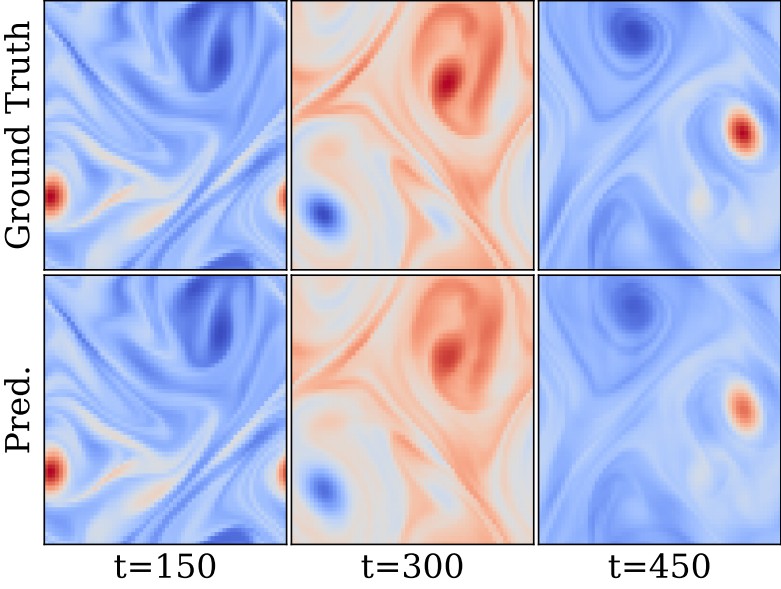

Figure 8: Sample of the ground truth sequence (**Top Row**) and predictions provided by NODA without data assimilation (**Bottom Row**) for the Navier-Stokes equation for $t \in \{150, 300, 450\}$.

