# OpenReview forum: "Learning semilinear neural operators: A unified recursive framework for prediction and data assimilation."
_ICLR.cc/2024/Conference — ICLR 2024 poster_

### Official Review · Reviewer_o6QX · 2023-10-19

**Soundness:** 2 fair
**Presentation:** 3 good
**Contribution:** 2 fair
**Rating:** 5
**Confidence:** 4

**Summary:**

The paper proposes the NODA method for learning semilinear neural operators that aloows both prediction and data assimilation. The method is motivated by the observer operator gain in [1], by which the evolution of the estimates are designed to consist of two terms, one for prediction and one for correction. Experiments on  the Kuramoto-Sivashinsky, NS, and Korteweg-de Vries equations are conducted to demonstrate the performance.

[1] Afshar, Sepideh, Fabian Germ, and Kirsten Morris. "Extended Kalman filter based observer design for semilinear infinite-dimensional systems." IEEE Transactions on Automatic Control (2023).

**Strengths:**

- The motivation is clear and applying data assimilation to neural operator learning is very relevant.
- The paper is well-written and easy-to-follow. The derivation of the method is intriguing.
- The integration of the observer design of semilinear PDEs seems novel.
- Strong experimental results.

**Weaknesses:**

- Lack of explanation of the observer design of semilinear PDEs. On page 5, the authors claim that under “mild” additional conditions, the solution converges to that of the real solution. In the experiments, the measurements are obtained by injecting Gaussian noises to the real solutions. It might be beneficial to provide some details of the claim so that readers may learn about in which data assimilation scenario this framework may fit.
- While the adoption of the FNO architecture is reasonable for the prediction term, it is not clear why the observer gain $K$ should/ are sufficient to be parametrized as in (15). A further explanation of the intuition here might be necessary.
- As for the experiments, it seems only the proposed NODA method needs a warmup. While I totally understand the methodology here and the need for the warmup, I just wonder if this would cause unfair comparison with other methods that have no access to this part of the data and thus less strong performance.
- It might be better to include colorbars for all the figures, not only for one in Figure 5.

**Questions:**

See weaknesses.

---

> ### Author Response · Authors · 2023-11-18
>
> We thank the Reviewer for the constructive suggestions which helped us to improve our paper. Regarding each of your suggestions:
> + **Explanation of observer design:**
>     Thank you for your suggestion, we have modified the paper in order to clarify the conditions for the observer design, which now reads:
>
>    "By solving a nonlinear infinite-dimensional Riccati equation, one can design an operator $K(t)$ based on the PDE in equation 10 that can compute the solution $\hat{z}(t)$ over time, such that under the absence of noise and disturbances, and under mild additional conditions, the solution satisfies $||z(t)-\hat{z}(t)||_{\mathcal{Z}_x}\to0$ as $t\to\infty$ for sufficiently small initial errors (see Theorem 5.1 in [Afshar2023]). Furthermore, the authors show that under bounded disturbances    $\omega(t)$ and $\eta(t)$, the estimation error can also be bounded for all $t$ (see Corollary 5.2 in [Afshar2023]). Similarly to the previous case, when $K(t)$ is strongly continuous the analytical form of a solution $\hat{z}(t)$ to (10) is given by:"
>
>    However, we emphasize that the proposed methodology is a fully data-driven framework, thus, the conditions for the proposed framework to work are not directly comparable to those in [Afshar2022].
>
>
> + **Intuition for the parametrization of observer gain:**
> To design the parametrization of the gain $K(t)$ as in Equation (15) we modified the gating intuition discussed in [Guen2020]. There, the gain works as a gate controlling the influence of the measurements in the updated states. When $K(t) = 0$ the measurement does not influence the updated states. Conversely, when $K(t)=I$,
> the latent dynamics is only driven by the measurements. However, this strategy does not accommodate the measurement  operators $C$ and so, we modified their design.
> In our scenario, $K$ corrects the predicted state based on the error in the predicted measurements, thus, better approximating the observer design. Note, however, that when $C = I$, the same gating interpretation holds for the gain design in (15) for $K\equiv I$.
> To clarify this intuition we added the comment "To parametrize the observer gain $K$, we generalized the gating architecture used in (Guen \& Thome, 2020) to cope with general measurement operators $C$ and the observer design:" when introducing (15) in the revised paper.
>
> + **Effect of warmup:**
>     We understand the Reviewer's point. Indeed, the proposed strategy uses the warmup data to improve its internal estimated states. Nevertheless, the presented experiments highlight the proposed strategy's benefits when compared with current state-of-the-art NO techniques. Also note that we performed experiments for different lengths of warmup periods $t_H$ (shown in Figure 3 of the main paper) which show that as the warmup time reduces the results of the proposed approach approach those of the standard FNO method. Nonetheless, our contribution is an observer-design-inspired framework that can be coped with different NO strategies besides FNO.
>
>
> + **Including colorbars on all figures:** We have included colorbars on all figures in the revised paper.
>
> An updated version of our paper incorporating the aforementioned changes will be uploaded by the end of the rebuttal period.
>
>
> [Guen2020] Vincent Le Guen and Nicolas Thome. Disentangling physical dynamics from unknown factors for unsupervised video prediction. In Proceedings of the IEEE/CVF Conference on Computer Vision and Pattern Recognition, pp. 11474–11484, 2020.

---

> > ### Comment · Reviewer_o6QX · 2023-11-22
> >
> > Thank you for addressing my comments and concerns.

---

> > > ### Author Response · Authors · 2023-11-22
> > > **Revised manuscript uploaded**
> > >
> > > Dear reviewer, we submitted the revised manuscript. We believe to have addressed most of the reviewer's points and hope the revised manuscript matches your expectations. If so, we kindly ask to increase the rating provided in the first revision round.

---

### Official Review · Reviewer_M6oo · 2023-10-30

**Soundness:** 3 good
**Presentation:** 3 good
**Contribution:** 3 good
**Rating:** 8
**Confidence:** 4

**Summary:**

Acknowledging the recent limitations in current Neural Operators, namely, data assimilation and lack of forecast error correction, authors propose learning semilinear operators instead, for which a body of mathematics literature exists already and allows for data assimilation and error correction through the use of the *observer* operator. Another feature of this framework is the fact that it assumes that the measurements are inherently noisy and also assumes in general that the dynamics themselves could be noisy. The authors base their architectural design on the semilinear operator by proposing a time-discretization that they later parametrize using Fourier Neural Operators (FNO). An essential facet of their research lies in their emphasis on learning correction operators, and this is substantiated by comprehensive ablation studies that illustrate their efficacy. In summary, the method presented in this work yields notably robust outcomes, particularly in the context of the considered datasets.

The first weakness that I pointed to in the weakness section is a huge one in my opinion and is reflected in my score. I am ready to change my score if that weakness is addressed since I think it will make the whole paper much better.

**Strengths:**

- Well-written paper and method is well motivated with respect to the literature.
- Considers the snapshots as inherently noisy instead of assuming their ground-truth nature.
- Proposed framework is baked in sound mathematical theory which is reflected in the design choice of the architecture, furthemore, it proposes a way of mitigating the long-standing challenge of long-term forecast error.
- Proposed framework beats the baselines when $\alpha=0$ suggesting that even in the prediction-only phase (Tables 1, 2 and 4), the architecture is well justified. Moreover, the authors demonstrate the soundness of the correction operator through (Table 3, Figures 3 and 4) in which they show that the error decreases as more updates steps are considered.

**Weaknesses:**

- Unclear how Eq. 12 was derived. I tried going through the effort of deriving it myself but couldn't get the same results, so it's critical that this step is justified (you can include the proof in the appendix if it burdens the main text) but a proof must be provided since the whole method is based on it.
- This is a "weak" weakness but given that the setting considered in the paper is that of $y(t)$ being just noise added to $z(t)$, it seems perhaps too much trying to learn a linear operator ($I_d$ in this case) using a non-linear neural net. While I understand that this was done to accomodate general case scenarios, it would be nice to see some ablation experiments showing the superiority of using a neural net for learning the operator $C$ as compared to using either: A learnable projection matrix or using not learning it at all and assuming $C=I_d$. Also, having an additional dataset where $C$ is not trivial could help motivate the choice of the architecture for it.
- In the main text and the paper from Afshar et al. (2022), The *observer gain* $K(t)$ needs to satisfy the condition that in the absence of noise from the system dynamics $z(t)$ and noise from the measurements $y(t)$, $\hat{z}(t)\rightarrow z(t)$, yet nothing is said about that when the architecture for it was introduced in Eq. 15. Perhaps a pretraining period where no noise is introduced in the samples could be used to make sure that the learned $K$ satisfies the aforementioned condition (at least approximately) and then finetuning further using your training scheme.

**Questions:**

- Is it possible to have a clear demonstration of why Navier-Stokes and KdV equations are semi-linear PDEs?

---

> ### Author Response · Authors · 2023-11-19
> **Response 1 of 2**
>
> We thank the Reviewer for the constructive suggestions to improve our paper. Regarding each of your suggestions:
> +  **Derivation of equation 12:** First, let us note that as explained in [Afshar2023] equation (11) is a solution to the observer system in equation (10) when $K(t)$ is strongly continuous, due to proposition 2.4 in [Afshar2023]. To improve clarity, we have made precise the need for the strong continuity of $K(t)$ for the solution in equation (11) to be valid right before equation (11) in the revised paper. Equation (12) consists of a time-discretization of equation (11). It can be derived by using the semigroup property of $T(t)$ [Curtain2020], which satisfies
>  \begin{align}
> T(t+t')=T(t)T(t')
> \end{align}
>  for any pair $t,t'$. This way, identifying $t$ in equation (11) with the discrete time $t_k$ we can express $\hat{z}(t_k)$ as:
>   \begin{align}
>     \hat{z}(t_k) ={}& T(t_k)\hat{z_0} + \int_{0}^{t_k}T(t_k-s)\Big[G(\hat{z}(s),s) + K(s)[y(s) - C\hat{z}(s)]\Big]ds
>     \\\\
>     ={}&  T(t_k-t_{k-1}+t_{k-1})\hat{z_0} + \int_{0}^{t_{k-1}}T(t_k-t_{k-1}+t_{k-1}-s)\Big[G(\hat{z}(s),s) + K(s)[y(s) - C\hat{z}(s)]\Big]ds
>     \\\\
>     +{}& \int_{t_{k-1}}^{t_k}T(t_k-s)\Big[G(\hat{z}(s),s) + K(s)[y(s) - C\hat{z}(s)]\Big]ds
>     \\\\
>     ={}& T(t_k-t_{k-1})T(t_{k-1})\hat{z_0} + T(t_k-t_{k-1})\int_{0}^{t_{k-1}}T(t_{k-1}-s)\Big[G(\hat{z}(s),s) + K(s)[y(s) - C\hat{z}(s)]\Big]ds
>      \\\\
>     +{}& \int_{t_{k-1}}^{t_k}T(t_k-s)\Big[G(\hat{z}(s),s) + K(s)[y(s) - C\hat{z}(s)]\Big]ds
>     \\\\
>     ={}& T(t_k-t_{k-1}) \underbrace{\bigg[T(t_{k-1})\hat{z_0} + \int_{0}^{t_{k-1}}T(t_{k-1}-s)\Big[G(\hat{z}(s),s) + K(s)[y(s) - C\hat{z}(s)]\Big]ds\bigg]}
>    \\\\
>    +{}& \int_{t_{k-1}}^{t_k}T(t_k-s)G(\hat{z}(s),s)ds + \int_{t_{k-1}}^{t_k}T(t_k-s)K(s)[y(s) - C\hat{z}(s)]ds \,
>   \end{align}
> where we used the linearity of $T(t)$. By noting that the term inside the large brackets highlighted in the previous equation is equal to
> \begin{align}
>   T(t_{k-1})\hat{z_0} + \int_{0}^{t_{k-1}}T(t_{k-1}-s)\Big[G(\hat{z}(s),s) + K(s)[y(s) - C\hat{z}(s)]\Big]ds=\hat{z}(t_{k-1})
> \end{align}
> it can be seen that this equation is now in the same form as equation (12). We have included this derivation as an appendix in the revised paper, and added a remark pointing to this derivation right before equation (12) in the main body of the paper.
> + **Experiments with non-identity operator:** We are running the experiments with NS with random full-rank $C$ matrices. We will include this extra experiment in an appendix.
> + **Conditions for observer design in Afshar's paper:** We agree with the reviewer that the observer design could be used to inform the architecture of the observer gain $K(t)$. However, we emphasize that the proposed approach is fully data-driven, and thus, the conditions for it to work are not directly comparable to those in [Afshar2022]. Moreover, during our experiments, we were able to obtain significant improvements in the experimental results compared to the competing techniques when using the parametrization of the observer gain in (15), without need for a noiseless pre-training period. Thus, although a pretraining step seems like an interesting way of tightening the connection between the optimal observer design and the proposed data-driven strategy, this will be the focus of followup work. We modified the text in the paper above equation (11) in the paper, where we discuss the conditions under which the optimal observer design is expected to work, which now reads:
>
>     "By solving a nonlinear infinite-dimensional Riccati equation, one can design an operator $K(t)$ based on the PDE in equation 10 that can compute the solution $\hat{z}(t)$ over time, such that under the absence of noise and disturbances, and under mild additional conditions, the solution satisfies $||z(t)-\hat{z}(t)||_{\mathcal{Z}_x}\to0$ as $t\to\infty$ for sufficiently small initial errors (see Theorem 5.1 in [Afshar2023]). Furthermore, the authors show that under bounded disturbances $\omega(t)$ and $\eta(t)$, the estimation error can also be bounded for all $t$ (see Corollary 5.2 in [Afshar2023]). Similarly to the previous case, when $K(t)$ is strongly continuous the analytical form of a solution $\hat{z}(t)$ to (10) is given by:''

---

> ### Author Response · Authors · 2023-11-19
> **Response 2 of 2**
>
> + **NS and KdV as semilinear PDEs:** For better convenience, let us recall here the expression of these PDEs, for the NS equation:
>     \begin{align}
>         \frac{\partial z(x,t)}{\partial t} + u(x,t)\nabla z(x,t) = v\Delta z(x,t) +f(x)
>     \end{align}
>     where $\nabla \cdot u(x,t)=0$, $z=\nabla \times u$;
>     and for the KdV equation, which we have write it into an easier-to-interpret form (compared to its form in the original paper) as:
>     \begin{equation}
>         \frac{\partial z}{\partial t} = -z\frac{\partial z}{\partial x}-\frac{\partial^3 z}{\partial^3 x}
>     \end{equation}
> In short, both the NS and KdV equations can be viewed as semilinear evolution PDEs as they contain 1) only a liner first order differential term involving the time variable, 2) linear differential terms that do not depend on the time derivatives of the solution $z$, and 3) nonlinear terms which also do not depend on the time derivatives of the solution $z$ and satisfy some regularity conditions.
> However, formally showing that this is the case is more laborious, thus we spare us both the work and point to some relevant references. For the NS equation, this can be found, for instance, in Section 3.3 of [Temam1982]. We have included this reference in the revised manuscript. For the KdV equation, which is known to be a third order semilinear PDE [Sonner2022], we have included a reference and also rewritten it using a clearer notation, as in the equation shown just above in this response, which we believe should make the connection to the general semilinear evolution PDE form easier for the reader to visualize.
>
> An updated version of our paper incorporating the aforementioned changes will be uploaded by the end of the rebuttal period.
>
> [Afshar2023] Sepideh Afshar, Fabian Germ, and Kirsten Morris. Extended Kalman filter based observer design for semilinear infinite-dimensional systems. IEEE Transactions on Automatic Control, 2023.
>
> [Curtain2020] Ruth Curtain and Hans Zwart. Introduction to infinite-dimensional systems theory: a state-space approach, volume 71. Springer Nature, 2020.
>
> [Temam1982] Roger Temam. Behaviour at time t=0 of the solutions of semi-linear evolution equations. Journal of Differential Equations, 43(1):73–92, 1982.
>
> [Sonner2022] Stefanie Sonner. An introduction to partial differential equations. Radboud University, Nijmegen, IMAPP - Mathematics, 2022

---

> > ### Comment · Reviewer_M6oo · 2023-11-21
> >
> > I thank the reviewers for addressing my comments and concerns. I believe with the aforementioned changes, the paper would be of higher quality and as a result, I am increasing my score to reflect that.

---

> > > ### Author Response · Authors · 2023-11-22
> > > **Revised manuscript uploaded**
> > >
> > > Dear reviewer, we submitted the revised manuscript. We thank the reviewer again for their suggestions that helped us improve the quality of our manuscript.

---

### Official Review · Reviewer_2zWc · 2023-11-04

**Soundness:** 1 poor
**Presentation:** 1 poor
**Contribution:** 1 poor
**Rating:** 5
**Confidence:** 3

**Summary:**

The paper introduces a learning-based state-space approach to address the challenges associated with solving complex systems governed by spatio-temporal Partial Differential Equations (PDEs) over long time scales. It highlights the limitations of existing Neural Operators (NOs) when dealing with data assimilation and correction operations based on noisy measurements. The proposed framework leverages the structure of semilinear PDEs and nonlinear observers to develop a recursive method that combines prediction and correction operations efficiently. The paper presents promising results from experiments on various PDEs, showcasing the model's robustness and ability to correct predictions with irregularly sampled noisy measurements.

**Strengths:**

1. **Innovative Approach**: The paper presents a novel learning-based state-space approach to address a critical issue in the context of solving spatio-temporal PDEs over long time scales, providing a fresh perspective on solving complex problems in science and engineering.

2. **Efficiency and Accuracy**: The proposed framework aims to produce fast and accurate predictions while handling irregularly sampled noisy measurements, enhancing the accuracy of PDE solutions.

3. **Real-World Relevance**: The need for data assimilation and correction in dynamical systems, as discussed in the paper, has significant real-world relevance, particularly in fields such as Earth science, remote sensing, traffic analysis, and medical imaging.

4. **Robustness to Noise**: The experiments demonstrate the model's robustness to noise, which is a crucial consideration in real-world applications where measurements are often affected by noise and uncertainties.

**Weaknesses:**

1. **Complexity**: While the proposed approach appears promising, the complexity of the model and the methods discussed could make implementation and practical application challenging for researchers and practitioners without expertise in this specific field.

2. **Limited Application Scenarios**: The paper primarily focuses on solving PDEs and addressing issues related to data assimilation and correction. It would be beneficial to discuss broader applications and practical scenarios where this approach could have a significant impact.

3. **Computational Overhead**: Although the paper suggests that the proposed model has little computational overhead, it would be valuable to provide more specific information regarding the computational resources required for practical implementation.

**Questions:**

1. Can you elaborate on the specific scenarios or application domains where the proposed learning-based state-space approach is expected to have the most significant impact?

2. How does the model handle variations in the amount of available measurements, and are there limitations in terms of the minimum number of measurements required for effective data assimilation?

3. Could you provide more details about the computational resources needed for implementing and running the proposed framework in practical applications, particularly in scenarios involving large-scale systems?

4. In the context of the experiments conducted, are there specific parameters or settings that were found to be critical for achieving the robustness of the model to noise?

5. Are there plans or ongoing research aimed at simplifying the implementation and improving the user-friendliness of the proposed approach for researchers and practitioners in related fields?

These questions aim to gain further insights into the practical applicability and potential impact of the proposed learning-based state-space approach for solving complex systems described by spatio-temporal PDEs.

---

> ### Author Response · Authors · 2023-11-18
>
> We thank the Reviewer for the constructive suggestions to improve our paper. Regarding each of your suggestions:
>
> + **Scenarios where the proposed approach can have the most impact:**
> We have discussions connecting NO literature with applications appearing in the introduction and related work sections of the manuscript. However, we understand that connecting the benefits of the data assimilation component of the proposed operator approximation framework is paramount. For this reason we added, just before stating the contributions of the paper in the introduction section, the following sentence:
>   "This can have significant impact in practical applications including, e.g., weather and Earth surface temperature forecast (Pathak et al., 2022; Jiang et al., 2023), remote sensing imaging (Weikmann et al., 2021; Borsoi et al., 2021) and fMRI dynamics (Singh et al., 2021; Buxton, 2013)."
>
> + **How the model handles variations in the amount of measurements:**
>    The influence of the amount of measurements, defined by $\alpha$, is an important question. This is evaluated in the experiments whose results are shown in Table 3 in the main body of the paper. Specifically, this experiment evaluates the performance of NODA ranging from the case of $\alpha=0$% (with no data for assimilation) up to the case $\alpha=30$% (where one every three noisy samples is used for assimilation). It can be seen that NODA performs well compared to the baselines even when no data is available for assimilation (i.e., $\alpha=0$%), and that its performance increases gradually with the amount of data available for assimilation (with a steady reduction of approximately $0.005$ in the $\text{RelMSE}$ estimation error for every increase of 10% in $\alpha$).
>
> + **Details about computational resources and computational overhead:** We would like to point the reviewer to Appendix B where we have provided the details of the computational resources used for conducting all the experiments.
>     To compare the NODA with the competing methods with respect to the computational overhead required, we are running simulations to evaluate the timings of the different methods, and will include them in the revised version of the paper by the rebuttal deadline.
>
> + **Critical parameters or settings for achieving the robustness of the model to noise:**
> In our practical experience the hyperparameters of NODA didn't play a huge role when considering noisy measurements. In fact we used the same hyperparameters across all experiments, which include both noiseless cases, as well as noisy data with different signal-to-noise ratios (20 and 30 decibels).
>
> + **User friendliness:**
>     Yes, we are indeed leveraging the recursive neural operator-based framework in an applied work that is being currently developed.
>     Furthermore, we will make the source code publicly available upon acceptance of the paper. We also highlight that although the mathematical content that serves as the foundation for the proposed framework can be fairly involved, the overall implementation of the NODA algorithm is fairly straightforward since it boils down to a sequence of prediction and update steps shown in equations (13) and (14) of the paper.
>
> An updated version of our paper incorporating the aforementioned changes will be uploaded by the end of the rebuttal period.

---

> > ### Author Response · Authors · 2023-11-22
> > **Revised manuscript uploaded**
> >
> > Dear reviewer, we submitted the revised manuscript. We believe to have addressed most of the reviewer's points and hope the revised manuscript matches your expectations. If so, we kindly ask to increase the rating provided in the first revision round.

---

### Official Review · Reviewer_ozZs · 2023-11-08

**Soundness:** 3 good
**Presentation:** 4 excellent
**Contribution:** 3 good
**Rating:** 8
**Confidence:** 4

**Summary:**

The paper presents NODA, a novel means of solving a particular class of PDEs with Neural Operators.
By exploiting the structure of semilinear PDEs, NODA benefits from increased accuracy on suitable tasks, and the important ability to incorporate noisy data to assist in making its predictions. Experiments are run on two different PDEs, Kuramoto-Sivashinsky and Navier-Stokes, in one and two spatial dimensions respectively.

**Strengths:**

Originality: NODA is, to the best of my knowledge, a highly novel approach to leveraging the structure of semilinear PDEs to create more powerful Neural Operators.

Motivation: The authors make a strong case for NODA's theoretical motivation, by leveraging the structure of semilinear PDEs, and introducing an interesting prediction/correction mechanism.

Clarity: The paper is well written, with a clear exposition of the concepts and methodology.

Significance: This is certainly a highly impactful area of machine learning, one which could lead to improvements in the speed and accuracy of predicting complex dynamical systems.

**Weaknesses:**

- The abstract claims that the proposed method makes fast predictions, and has “little computational overhead”. Yet no experimental evidence is provided to support this statement. It’s important the authors rectify this by including quantitative timing metrics for the experiments, and compare it against the other methods presented.

- The selection of experiments is very limited – only looking at one 1D and one 2D PDE. It would be advisable to at least present results for more than one Reynolds number.

- The  experimental results are well presented, but the tables lack uncertainty estimates. The reader is therefore left unable to gauge whether the performance differences between different methods are statistically significant.

- While there are a good number of alternative methods in the benchmarks, I’m not sure they were the most appropriate choices. The two LSTM-based models are taken from papers that are over eight years old. Meanwhile the MNO is specifically designed for dissipative dynamics, yet the Navier-Stokes experiment is chosen to be in the non-dissipative regime. Furthermore, none of the other methods selected for benchmarking are able to make use of the noisy data, so we are unable to test the key feature of NODA. Alternative works which may offer more suitable benchmarks include ‘Approximate Bayesian Neural Operators’ and ‘Multiwavelet-based Operator Learning for Differential Equations’.

**Questions:**

My suggestions are linked to the weaknesses highlighted above:

- Including timings for the experiments, to show how fast the algorithm is compared to competing methods.

- To introduce more variety in the experiments. Would it be impractical to explore higher Reynolds numbers, and do we expect NODA to cope less well with turbulent flow?

- I imagine the uncertainties in the numerical experiments are quite small but it would still be a strong recommendation to include them.

- Either some justification needs to be added for the particular choices of the benchmark methods, or more suitable alternatives should be introduced in their place.

- Figures 1 and 2 are lacking a colour scale, making it challenging for the reader to assess the nature of the uncertainties. The reader is left to guess values based upon the Table but it would be preferable to be explicit and quantitative.

Overall I would stress that I think the paper is very promising, so I hope these suggestions help the authors to address my concerns.

---

> ### Author Response · Authors · 2023-11-18
>
> We thank the Reviewer for the constructive suggestions which helped us to improve our paper. Regarding each of your suggestions:
>
> + **Including timings for the experiments** We are running the simulations to evaluate the timings, and will include them in the revised version of the paper.
>
>
> +  **Including uncertainties in the experimental results:**
>     We thank the Reviewer for pointing that out. We added the uncertainties to our tables, and they are small compared with the performance gain obtained with NODA.
>
> + **Including more variety in the experiments:** We note that an extra experiment in the Kortweg-De Vries (KdV) equation is present in the appendix of the submitted manuscript. Nonetheless, we are working on additional experiments for NS with $R_e\in$ {500, 5000} for FNO, MNO and NODA that will be included in the appendix of the revised manuscript by the rebuttal deadline.
>
>
> + **Justification for the benchmark methods:** To the best of our knowledge, NODA is the first method to integrate NO learning, data assimilation, and a flexible system modeling that can also account for, e.g., measurement noise. Thus, it is indeed difficult to find baseline methods that would allow for a perfectly fair comparison.
> We have selected MNO and LSTM-based methods due to their recursive structure, which resembles one of the main aspects of our NODA, while FNO is used as a basic block in the architecture of NODA.
> We have clarified the justification for the selected baselines in the beginning of Section 4 in the revised paper, which reads:
>
>   "The MNO and LSTM-based methods were used as baselines due to their recurrent nature, which is also one of the main aspects of NODA, while FNO was selected since it is one of the main building blocks used in the NODA architecture."
>
>   The baselines proposed by the Reviewer are definitely relevant, and we have included them in the related work section of the revised paper. We are currently working to evaluate them on the considered datasets, but, unfortunately, due to limited time and computational resources we might not be able to include them before the end of the rebuttal period. Nevertheless, in recognizing the benefit of such as comparison, we are working on obtaining results to compare NODA with the Multiwavelet NO approach [Gupta2021].  Regarding the Bayesian approach proposed in [Magnani2022], we could not find its source code in online open repositories. For this reason, we only discuss it in the related work section.
>
>
>
>
> + **Including colorbars in the figures:** We have included colorbars in all figures in the revised paper.
>
>
> An updated version of our paper incorporating the aforementioned changes will be uploaded by the end of the rebuttal period.
>
> [Gupta2021] Gaurav Gupta, Xiongye Xiao, and Paul Bogdan. Multiwavelet-based operator learning for differential equations. Advances in neural information processing systems, 34:24048–24062, 2021.
>
> [Magnani2022] Emilia Magnani, Nicholas Kramer, Runa Eschenhagen, Lorenzo Rosasco, and Philipp Hennig. Approximate Bayesian neural operators: Uncertainty quantification for parametric PDEs. arXiv preprint arXiv:2208.01565, 2022.

---

> > ### Comment · Reviewer_ozZs · 2023-11-22
> >
> > Thank you for your thoughtful response, I look forward to reading the updated version.

---

> ### Author Response · Authors · 2023-11-22
> **Revised manuscript uploaded**
>
> Dear reviewer, we submitted the revised manuscript. In particular, we were also able to include the approach of [Gupta2021] as a benchmark method for the NS and KdV examples in our experiments: in brief, its performance stood between that of FNO and that of the proposed NODA. We believe to have addressed most of the reviewer's points and hope the revised manuscript matches your expectations. If so, we kindly ask to increase the rating provided in the first revision round.

---

> > ### Comment · Reviewer_ozZs · 2023-12-05
> >
> > I have read through the revised manuscript and was pleased to see the authors have made substantial improvements. I have adjusted my score accordingly.

---

### Meta-Review · Area_Chair_osrv · 2023-12-06

**Metareview:**

This paper provides a novel variant of neural operators tailored to semi-linear PDEs, and also offers novel contributions to data assimilation in such settings. Authors and reviewers had fruitful discussions, to a point where the majority of reviewers now agree that the paper is valuable, novel, and well presented. It should be accepted.

I also want to note that review `2zWc` bears multiple features that indicate the use of an LLM, including the unusual score distribution, and overall phrasing. This reviewer also did not engage in the discussion. I have thus ignored this review in my evaluation.

**Justification For Why Not Higher Score:**

I wouldn't mind bumping it up. But semi-linear PDEs are not a novel concept, in fact they are a widely leverage structure in classic PDE solvers. As such the paper, while novel and valuable in the neural operator context, is not a major conceptual advancement in my opinion.

**Justification For Why Not Lower Score:**

It's definitely a well-written, convincing, interesting paper.

---

### Decision · Program_Chairs · 2024-01-16

Accept (poster)